# Ionic Liquid-Assisted Synthesis of Ag$_3$PO$_4$ Spheres for Boosting Photodegradation Activity under Visible Light

**Beibei Zhang [1], Lu Zhang [2], Yulong Zhang [3], Chao Liu [3,\*], Jiexiang Xia [2] and Huaming Li [2]**

1   School of Chemistry & Chemical Engineering, Yancheng Institute of Technology, Yancheng 224051, China; yizhou1118@163.com
2   School of Chemistry and Chemical Engineering, Institution for Energy Research, Jiangsu University, Zhenjiang 212013, China; luzhang0006@163.com (L.Z.); xjx@ujs.edu.cn (J.X.); lihm@ujs.edu.cn (H.L.)
3   School of Materials Science and Engineering, Yancheng Institute of Technology, Yancheng 224051, China; zylacj@hotmail.com
\*   Correspondence: cliu@ycit.edu.cn; Tel.: (+86)-515-88298270

**Abstract:** In this work, a simple chemical precipitation method was employed to prepare spherical-like Ag$_3$PO$_4$ material (IL-Ag$_3$PO$_4$) with exposed {111} facet in the presence of reactive ionic liquid 1-butyl-3-methylimidazole dihydrogen phosphate ([Omim]H$_2$PO$_4$). The crystal structure, microstructure, optical properties, and visible-light photocatalytic performance of as-prepared materials were studied in detail. The addition of ionic liquids played a crucial role in forming spherical-like morphology of IL-Ag$_3$PO$_4$ sample. Compared with traditional Ag$_3$PO$_4$ material, the intensity ratio of {222}/{200} facets in XRD pattern of IL-Ag$_3$PO$_4$ was significantly enhanced, indicating the main {111} facets exposed on the surface of IL-Ag$_3$PO$_4$ sample. The presence of exposed {111} facet was advantageous for facilitating the charge carrier transfer and separation. The light-harvesting capacity of IL-Ag$_3$PO$_4$ was larger than that of Ag$_3$PO$_4$. The photocatalytic activity of samples was evaluated by degrading rhodamine B (RhB) and p-chlorophenol (4-CP) under visible light. The photodegradation efficiencies of IL-Ag$_3$PO$_4$ were 1.94 and 2.45 times higher than that of Ag$_3$PO$_4$ for RhB and 4-CP removal, respectively, attributing to a synergy from the exposed {111} facet and enhanced photoabsorption. Based on active species capturing experiments, holes (h$^+$), and superoxide radical ($\bullet$O$_2{}^-$) were the main active species for visible-light-driven RhB photodegradation. This study will provide a promising prospect for designing and synthesizing ionic liquid-assisted photocatalysts with a high efficiency.

**Keywords:** photocatalysis; ionic liquid; degradation; Ag$_3$PO$_4$; synergistic effects

## 1. Introduction

Environmental photocatalysis is a low-energy, green, and sustainable advanced oxidation technology. Under light irradiation, semiconductor materials can be excited to generate electron-hole pairs and then produce oxygen-containing free radical species with a strong oxidizing ability for completely mineralizing organic pollutants [1–3]. As a representatively traditional photocatalyst, titanium dioxide (TiO$_2$) shows the intrinsic characteristics of narrow light response range, high charge carrier recombination rate, and low quantum efficiency, which greatly limits its development and practical application in photocatalytic field [4]. Therefore, it is necessary to develop visible-light responsive semiconductor materials with a high efficiency for photocatalytic pollutant degradation, such as silver phosphate (Ag$_3$PO$_4$) [5], graphite carbon nitride (g-C$_3$N$_4$) [6], bismuth tungstate (Bi$_2$WO$_6$) [7], bismuth vanadate (BiVO$_4$) [8], and so on.

Among these reported visible-light photocatalysts, Ag$_3$PO$_4$ material, first prepared by Ye's group in 2010 [9], has been widely studied due to its strong photoabsorption ability in visible light region (<530 nm), narrow band gap (~2.4 eV) and extremely high photooxidation ability for decomposing water and organic dyes [10,11]. Nevertheless, the

Ag+ in $Ag_3PO_4$ is slightly soluble in solution and can be easily reduced to generate Ag (0) particles under light irradiation, resulting in the decreased photocatalytic activity. Recently, many efforts have been made to improve the photocatalytic activity of $Ag_3PO_4$, including crystal surface regulation [12], morphology design [13], heterojunction construction [14], etc. For crystal surface regulation of $Ag_3PO_4$, the surface energy of the exposed {111} facet was obviously higher than that of {110} or {100} facets [15]. The presence of {111} facets in $Ag_3PO_4$ crystal has a higher barrier, which could hinder the electron transfer and the charge carrier recombination, resulting in the efficient mobility/separation of charge carrier on the facets and thus the highly photocatalytic performance [16–18]. For example, the tetrahedral $Ag_3PO_4$ crystals composed of {111} facets exhibited extremely high water photooxidation activity, corresponding to the enhanced carrier mobility and increased active reactive sites resulted from exposed {111} facets. Thus, constructing $Ag_3PO_4$-based catalyst with exposed {111} facets have been proven to be an effective way to improve the photocatalyst activity.

The choice of phosphate groups is a critical role in controllably preparing $Ag_3PO_4$ with different morphologies and photocatalytic performances. Generally, the most studied phosphates mainly included $Na_3PO_4$, $Na_2HPO_4$ and $NaH_2PO_4$ [19,20]. As a new type of green medium, ionic liquids show some characteristics of high thermal stability, wide temperature range, low interfacial tension, and high ionic conductivity [21]. Ionic liquid-assisted synthesis of inorganic materials has been widely used and applied in photocatalytic fields, such as BiOI hollow microspheres for RhB photodegradation [22], porous perovskite-like $PbBiO_2Br$ for organic pollutants photodegradation [23], and porous sphere-like BiOBr hollow for RhB photodegradation [24]. Therefore, ionic liquids have certain significance in the controllable synthesis of $Ag_3PO_4$ materials. To the best of our knowledge, ionic liquid-assisted synthesis of $Ag_3PO_4$ material for photodegradation has not yet been reported.

In this work, IL-$Ag_3PO_4$ sample with exposed {111} facets have been synthesized via a chemical precipitation method using silver nitrate ($AgNO_3$) and [Omim]$H_2PO_4$ as the starting materials. The resulted samples were fully characterized. It was found that the presence of ionic liquid determined the microstructure and crystal surface of IL-$Ag_3PO_4$. The photocatalytic performances were evaluated by degrading 4-CP and RhB molecules under visible light irradiation, respectively. The trapping experiment was performed to confirm the main active species during RhB photodegradation. Based on the experimental results, a possible mechanism for improving photodegradation efficiency was proposed.

## 2. Results and Discussion

### 2.1. XRD Analysis

To confirm the crystal structure, the typical XRD patterns of as-prepared $Ag_3PO_4$ and IL-$Ag_3PO_4$ samples were shown in Figure 1. The characteristic diffraction peaks of (110), (200), (210), (211), (220), (310), (222), (320), (321) and (400) can be clearly visible for both $Ag_3PO_4$ and IL-$Ag_3PO_4$ samples, which perfectly index as the cubic phase of $Ag_3PO_4$ (JCPDS Card: No. 06-0505) [25]. No additional characteristic diffraction peaks, such as Ag or $Ag_2O$, are observed in $Ag_3PO_4$ and IL-$Ag_3PO_4$ samples, confirming the high purity of these two $Ag_3PO_4$ samples. However, the intensity ratios among the diffraction peaks in IL-$Ag_3PO_4$ are significantly changed related to $Ag_3PO_4$, which is ascribed to the corresponding ratio changes of exposed crystal facet [11]. Previous literature reported that $Ag_3PO_4$ crystal dominated by exposed {111} facets generally exhibited the best photocatalytic activity, because the higher surface energy generated by {111} facets was advantageous for facilitating the mobility and separation of charge carrier on the facets, thereby obtaining the highly photocatalytic performance [16,17]. Additionally, the presence of {111} facets, with a higher barrier, in $Ag_3PO_4$ crystal could also hinder the electron transfer and thus accelerate the recombination of charge carriers [18]. In this work, an intensity ratio of {222}/{200} facets of IL-$Ag_3PO_4$ (1.06) are significantly higher than that of $Ag_3PO_4$ crystal (0.83), which showed that the surface of IL-$Ag_3PO_4$ crystal is mainly {111} facets. The formation mechanism of exposed {111} facets is due to the fact

that the addition of ionic liquid can lead to the aggregation behavior. Then the ionic liquid effectively interacted with the crystal nuclear surfaces, which energetically distinguished different facets and promoted the growth of certain {111} facets [26]. It should be mentioned that the direct evidence of HRTEM images of IL-Ag$_3$PO$_4$ sample for observing exposed {111} facets cannot be provided due to the fact that the extremely large particle size of IL-Ag$_3$PO$_4$ sample cannot be measured under the current condition.

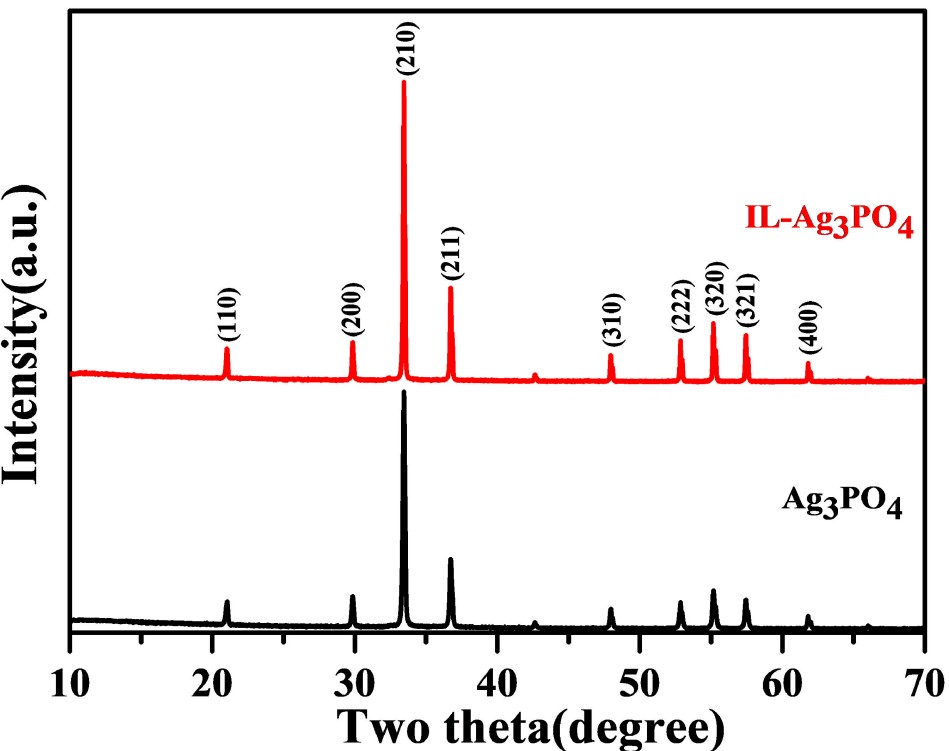

**Figure 1.** XRD patterns of Ag$_3$PO$_4$ and IL-Ag$_3$PO$_4$ photocatalysts.

*2.2. SEM Analysis*

The SEM images were shown in Figure 2 to achieve the morphology information of Ag$_3$PO$_4$ and IL-Ag$_3$PO$_4$ samples. From SEM images in Figure 2a,b, the sizes of traditional Ag$_3$PO$_4$ particles, with an irregular shape, vary from tens of nanometers to a few micrometers. After selecting ionic liquid [Omim]H$_2$PO$_4$ as the source of phosphate group, the SEM images of the resulted IL-Ag$_3$PO$_4$ in Figure 2c,d show a sphere-like morphology and the profoundly increased sizes of approximately 1~2.5 μm in comparison with Ag$_3$PO$_4$. Due to the presence of ionic liquids, the prepared materials may tend to form spherical particles with a small aggregation [27]. Obviously, the roles of ionic liquid ([Omim]H$_2$PO$_4$) can be used not only as the source of phosphate group but also a good dispersant/template agent to drive the formation of sphere-like morphology of Ag$_3$PO$_4$. Thus, the choice of ionic liquid with the cationic carbon chains is beneficial for forming uniform size nanomaterials and performing their surface regulation [28,29].

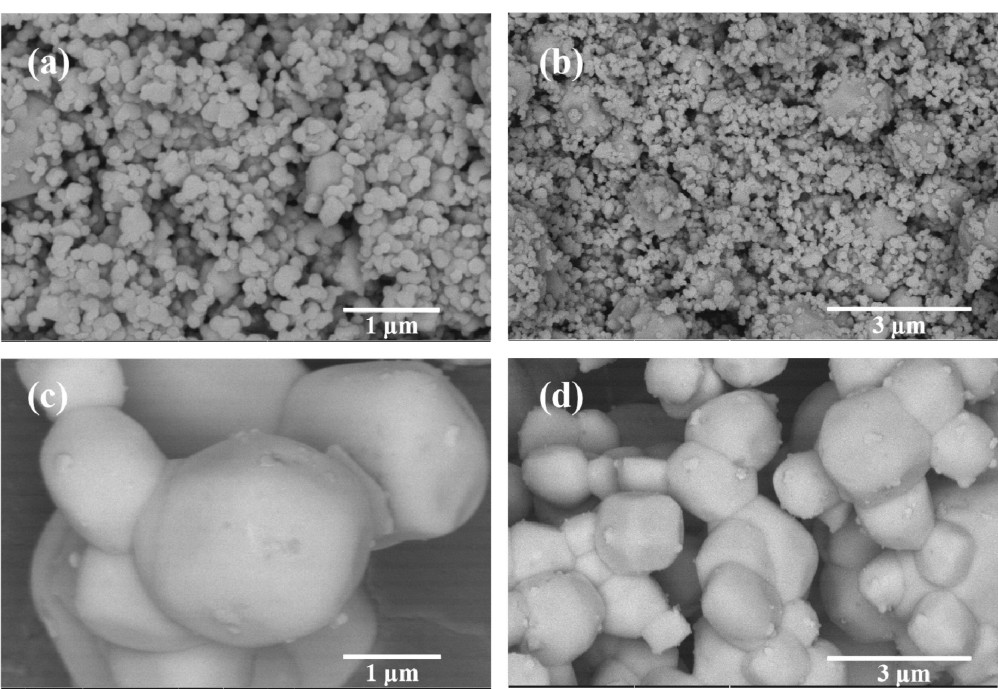

**Figure 2.** SEM images of (**a,b**) $Ag_3PO_4$ and (**c,d**) IL-$Ag_3PO_4$.

### 2.3. DRS Analysis

The optical properties of $Ag_3PO_4$ and IL-$Ag_3PO_4$ samples were evaluated by UV-vis DRS. As displayed in Figure 3a, the sample of $Ag_3PO_4$ exhibits an absorption edge at ~515 nm. When ionic liquid [Omim]$H_2PO_4$ was selected as the source of phosphate group, the absorption edge of IL-$Ag_3PO_4$ sample shows a slight red shift to around 530 nm. Based on these absorption edges, the bandgap values are estimated to be approximately 2.41 eV for $Ag_3PO_4$ and 2.34 eV for IL-$Ag_3PO_4$ [30]. Furthermore, the bandgap values of $Ag_3PO_4$ and IL-$Ag_3PO_4$ samples can be also calculated via a plot of $(\alpha h\nu)^{1/2}$ versus $h\nu$. The bandgap energies of $Ag_3PO_4$ and IL-$Ag_3PO_4$ in Figure 3b are estimated to be 2.46 and 2.40 eV, respectively, which are almost consistent with the bandgap values calculated by absorption edges. It indicates that the light-harvesting capacity of IL-$Ag_3PO_4$ is enhanced compared with $Ag_3PO_4$.

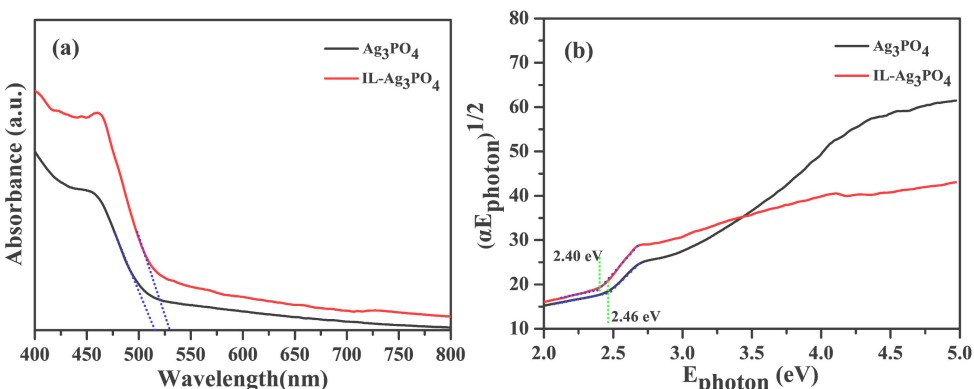

**Figure 3.** (**a**) UV-visible diffuse reflectance spectra (UV-vis DRS), and (**b**) the corresponding $(\alpha h\nu)^{1/2}$ vs. $h\nu$ plots of $Ag_3PO_4$ and IL-$Ag_3PO_4$.

### 2.4. XPS Analysis

The chemical composition and atomic state of $Ag_3PO_4$ and IL-$Ag_3PO_4$ samples were carried out using XPS measurement. As shown in Figure 4a, all peak positions can be

ascribed to Ag, P, O and C elements without any other impurities. The appeared C 1s peak is mainly attributed to the adventitious hydrocarbon from the XPS instrument itself. The high-resolution XPS spectra of Ag 3d in Figure 4b exhibits two peaks around 368.0 and 374.0 eV, which are ascribed to the binding energies of Ag 3d5/2 and Ag 3d3/2 of Ag$^+$ over Ag$_3$PO$_4$ and IL-Ag$_3$PO$_4$ samples [31]. The binding energy position of IL-Ag$_3$PO$_4$ is almost unchanged in comparison to Ag$_3$PO$_4$, indicating the similar chemical states of Ag$^+$.

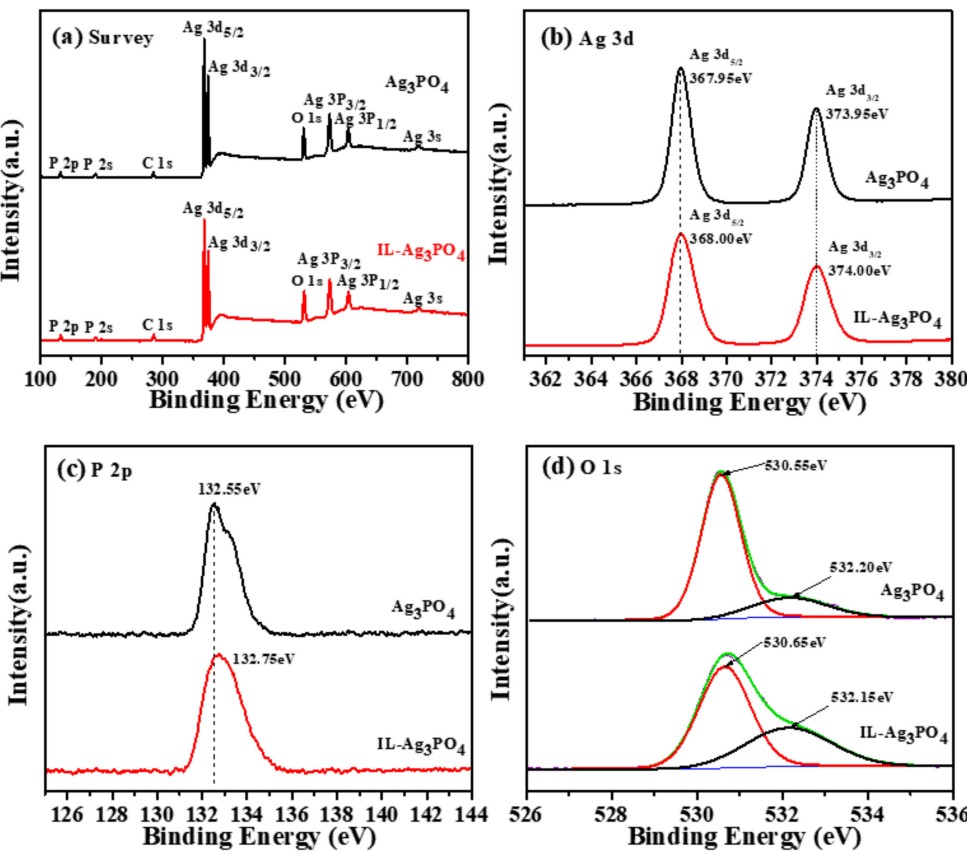

**Figure 4.** High resolution XPS spectra of (**a**) survey, (**b**) Ag 3d, (**c**) P 2p, and (**d**) O 1s.

The binding energy of P 2p at 132.75 eV in IL-Ag$_3$PO$_4$ sample (Figure 4c), attributed to the crystal lattice of P$^{5+}$ [32], is slightly shifted compared with Ag$_3$PO$_4$. As shown in Figure 4d, two peaks of O 1s around 532.15 and 530.65 eV are derived from hydroxyl groups on the surface and crystal oxygen in Ag$_3$PO$_4$, respectively [33]. It should be noted that the divided area in IL-Ag$_3$PO$_4$ centered at 532.15 resulted from surfaced hydroxyl groups is larger than that in Ag$_3$PO$_4$. This result illustrates that the sources of phosphate groups for preparing Ag$_3$PO$_4$ samples have some certain effects on the surface chemical environment.

## 2.5. Photocatalytic Activity

The photocatalytic activities of Ag$_3$PO$_4$ and IL-Ag$_3$PO$_4$ samples were evaluated by photodegradation using 4-CP and RhB as model pollutants under visible light [34]. As can be seen from Figure 5a,c, no obvious changes can be visible for pure 4-CP and RhB molecules in the absence of catalyst, indicating the relatively high structural stability. The adsorption of RhB and 4-CP molecules can reach equilibrium within 30 min. The photodegradation rate of 4-CP in IL-Ag$_3$PO$_4$ reach up to ~98.4% within 180 min, which is 2.45 times higher than that of Ag$_3$PO$_4$ (40.5%). The visible-light-driven photodegradation process curves of 4-CP over IL-Ag$_3$PO$_4$ were also investigated by HPLC as shown in Figure 5b. With the increase of irradiation time, the characteristic peaks of 4-CP with a retention time at 4.05 min are gradually decreased and almost disappeared within 180 min,

indicating the high mineralization rate. However, HPLC chromatogram shows two new peaks at the retention time of 1.62 and 2.67 min. As the process proceeds, the intensities of these two peaks have a gradual upward trend. It reveals that 4-CP molecules can be effectively degraded while a small number of intermediates are difficult to be fully removed by the photodegradation reaction.

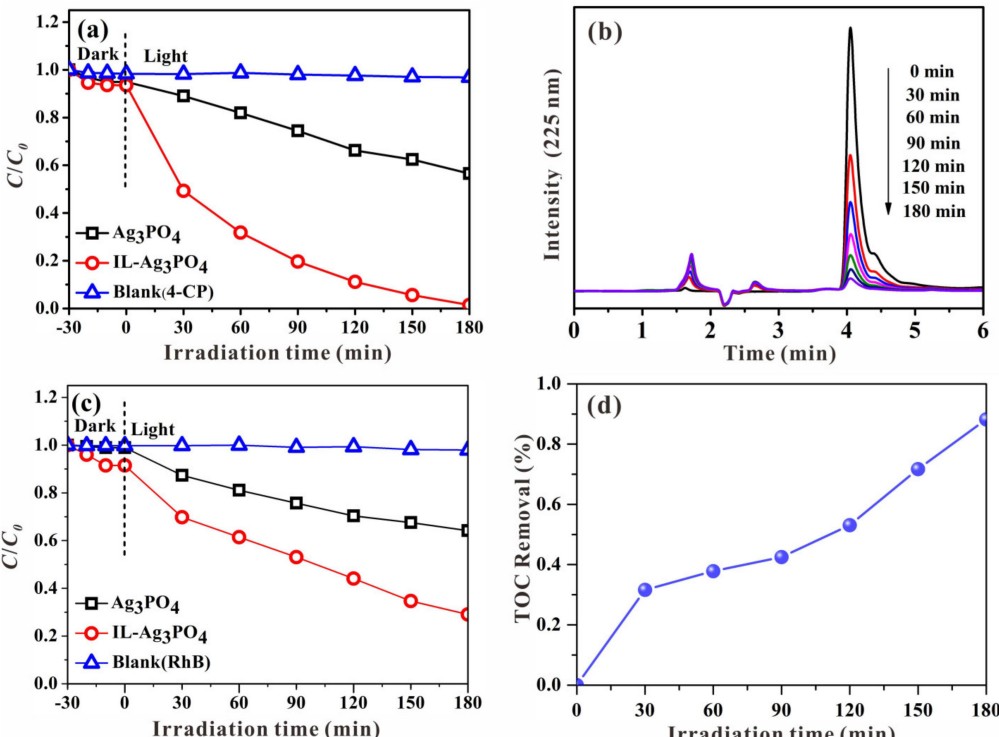

**Figure 5.** Photocatalytic degradation rate of (**a**) 4-CP and (**c**) rhodamine B (RhB) over different samples under visible light irradiation; (**b**) HPLC spectra and (**d**) TOC removal (relative to $C_0$) for photodegradation of 4-CP over IL-Ag$_3$PO$_4$ under visible light.

To further confirm the excellent activity of IL-Ag$_3$PO$_4$, the photocatalytic performance was also measured by the degradation of RhB molecules under visible light (Figure 5c). The RhB photodegradation rate of IL-Ag$_3$PO$_4$ is 68.23%, while only 35.13% for Ag$_3$PO$_4$ sample within 180 min. The photocatalytic efficiency of IL-Ag$_3$PO$_4$ is almost two-fold higher than that of Ag$_3$PO$_4$. Based on the above results, the high photodegradation efficiencies over IL-Ag$_3$PO$_4$ sample are assigned to the synergistic effects of the exposed {111} facets and enhanced photoabsorption. Furthermore, the TOC removal efficiency over IL-Ag$_3$PO$_4$ can reach approximately 88.2% within 180 min under visible light, demonstrating a high mineralization degree for 4-CP photodegradation.

### 2.6. Photocatalytic Mechanism Study

In order to clarify the photodegradation mechanism in IL-Ag$_3$PO$_4$ photocatalyst, it is necessary to determine the main active species [35]. Therefore, the IL-Ag$_3$PO$_4$ sample was carried out to capture the active radical species and the results was shown in Figure 6a. The added compounds of EDTA-2NA, p-benzoquinone and tert-butanol (*t*-BuOH) were served as scavengers of h$^+$, superoxide ($\bullet O_2^-$) and hydroxyl ($\bullet OH$) radicals, respectively. The RhB photodegradation efficiency is significantly reduced by the addition of p-benzoquinone and EDTA-2NA while no changes for the addition of *t*-BuOH, indicating that the active species of photogenerated h$^+$ and $\bullet O_2^-$ play a vital role in removing RhB molecule under visible light over IL-Ag$_3$PO$_4$ photocatalyst. This conclusion is consistent with the previous literature [36].

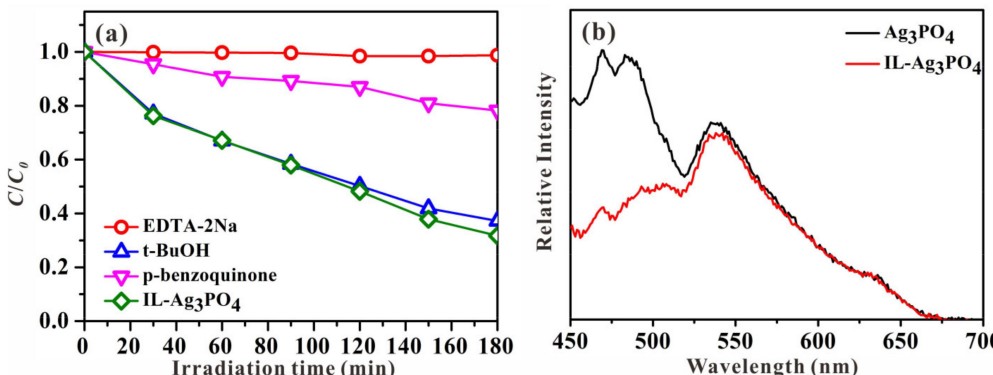

**Figure 6.** (**a**) Photocatalytic activities of the IL-$Ag_3PO_4$ for the degradation of RhB in the presence of different scavengers; (**b**) photoluminescence (PL) spectra of $Ag_3PO_4$ and IL-$Ag_3PO_4$.

The PL spectra of $Ag_3PO_4$ and IL-$Ag_3PO_4$ samples in Figure 6b were measured to evaluate the charge carriers trapping and recombination processes [37,38]. The higher PL intensity means the higher recombination rate of photoinduced charges. For $Ag_3PO_4$ sample, a strong emission peak at approximately ~540 nm can be clearly observed under the excitation of 381 nm. For comparison, the emission peak intensity of IL-$Ag_3PO_4$ material is lower than that of $Ag_3PO_4$ under the same measurement condition, demonstrating that the recombination probability of photogenerated charge carriers is restrained in IL-$Ag_3PO_4$ sample.

The electrochemical impedance spectra (EIS) and photocurrent response measurements were used to investigate the separation and transfer of photoinduced electrons and holes in both $Ag_3PO_4$ and IL-$Ag_3PO_4$ samples [39,40]. Figure 7a shows the EIS diagram of $Ag_3PO_4$ and IL-$Ag_3PO_4$ samples. The IL-$Ag_3PO_4$ sample exhibits the smaller arc radius than $Ag_3PO_4$, indicating the efficient charge transfer and separation. Furthermore, the photocurrent response of IL-$Ag_3PO_4$ (1.86 mA/cm$^2$) is 4.89 times higher than that of $Ag_3PO_4$ (0.38 mA/cm$^2$) in Figure 7b, indicating the effective separation of photogenerated charge carriers over IL-$Ag_3PO_4$ sample. The above results confirm that the presence of exposed {111} facets in IL-$Ag_3PO_4$ sample is advantageous for reducing recombination rate, and facilitating the transfer and separation of charge carrier, resulting in the highly photocatalytic efficiency.

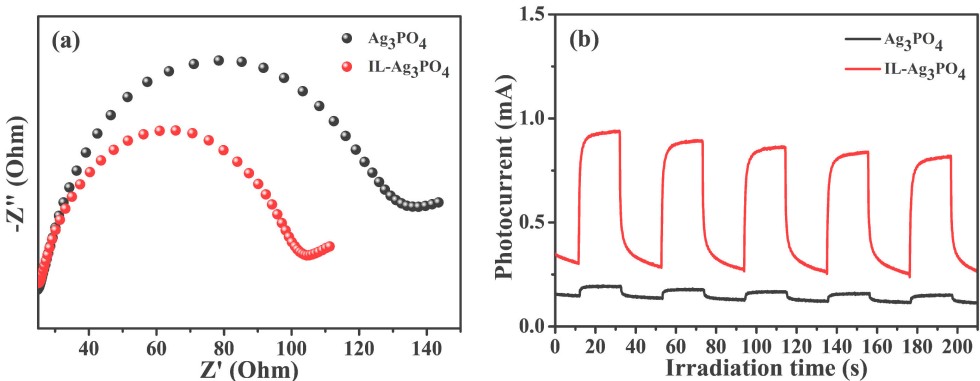

**Figure 7.** (**a**) Electrochemical impedance spectra and (**b**) transient photocurrent response of $Ag_3PO_4$ and IL-$Ag_3PO_4$.

Based on XPS valence band (VB) spectra in Figure 8a, the VB potential ($E_{VB}$) of IL-$Ag_3PO_4$ and $Ag_3PO_4$ can be estimated to be 2.58 and 2.62 eV. The above UV-vis DRS results demonstrate that the $E_g$ values of IL-$Ag_3PO_4$ and $Ag_3PO_4$ samples are 2.40 and 2.46 eV. According to formula of $E_g = E_{VB} - E_{CB}$, the minimum conduction band potential ($E_{CB}$) of IL-$Ag_3PO_4$ and $Ag_3PO_4$ are calculated to be 0.18 and 0.16 eV, respectively.

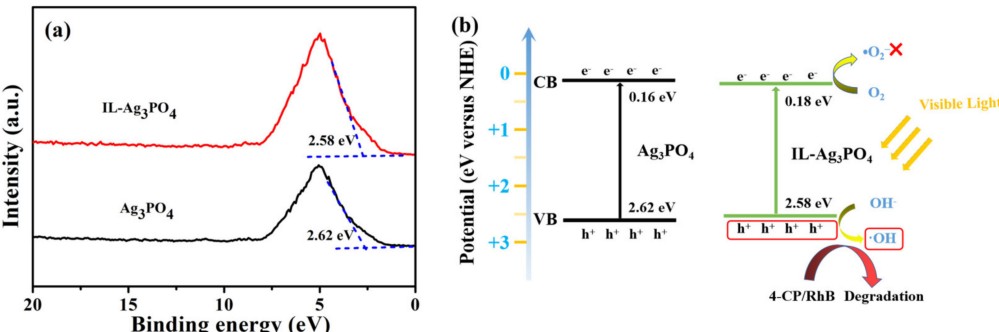

**Figure 8.** (**a**) XPS-valence spectra of $Ag_3PO_4$ and IL-$Ag_3PO_4$; (**b**) Schematic illustration of possible photocatalytic mechanism over IL-$Ag_3PO_4$ under visible light.

A possible visible-light photodegradation mechanism over IL-$Ag_3PO_4$ and $Ag_3PO_4$ was shown in Figure 8b. Compared with traditional $Ag_3PO_4$, the IL-$Ag_3PO_4$ sample exhibited the upward shift of VB position and the downward shift of CB position. It indicates the reduced bandgap value over IL-$Ag_3PO_4$ sample, leading to the enhanced light-harvesting capacity and thus the improved photodegradation activity [41]. Under visible light irradiation, the IL-$Ag_3PO_4$ semiconductor was excited to generate electrons and holes. Due to the dominant surface of {111}, the formed holes with strong oxidation ability rapidly transfer to the surface of the IL-$Ag_3PO_4$ photocatalyst, which have oxidation or reduction reactions with RhB on the catalyst surface to finally achieve degrading of pollutants [16]. The resulting electrons can flow into the conduction band and react with the $O_2$ to produce $\bullet O^{2-}$, which is responsive for RhB photodegradation. Therefore, the active species of holes and $\bullet O^{2-}$ play a vital role in removing RhB molecules under visible light over IL-$Ag_3PO_4$ photocatalyst.

## 3. Experimental Section

### 3.1. Materials

The silver nitrate ($AgNO_3$), sodium phosphate tribasic dodecahydrate ($Na_3PO_4 \cdot 12H_2O$), rhodamine B (RhB) and p-chlorophenol (4-CP) were purchased from the Sinopharm Group Chemical Co., Ltd. (Shanghai, China). The ionic liquid 1-butyl-3-methylimidazole dihydrogen phosphate ([Omim]$H_2PO_4$) was purchased from Shanghai Chengjie Chemical Co., Ltd. (Shanghai, China). All chemicals used in this study were of analytical grade without further purification.

### 3.2. Ionic Liquid-Assisted Synthesis of $Ag_3PO_4$ Spheres

The 0.340 g $AgNO_3$ was well dissolved in 20 mL deionized water and then 20 mL ionic liquid of [Omim]$H_2PO_4$ (0.1576 g) solution was added dropwise under a magnetic stirring. After heating at 60 °C for 1 h under dark condition, the solid product was collected by centrifugation, washed with distilled water and absolute ethanol several times, and then dried at 55 °C in a vacuum oven overnight. The resulted ionic liquid-induced $Ag_3PO_4$ was denoted as IL-$Ag_3PO_4$.

In order to investigate the role of the ionic liquid, the inorganic salt of $Na_3PO_4 \cdot 12H_2O$ was also selected as the source of phosphate group. The traditional $Ag_3PO_4$ was prepared using the same synthetic process as IL-$Ag_3PO_4$ except for the substitution of $Na_3PO_4 \cdot 12H_2O$ for ionic liquid [Omim]$H_2PO_4$.

### 3.3. Characterization of Photocatalysts

X-ray diffraction (XRD) patterns were obtained on a Bruker D8 diffractometer with Cu K$\alpha$ radiation ($\lambda$ = 1.5418 Å) and 2θ range from 10° to 70°. X-ray photoelectron spectroscopy (XPS) analysis was performed by using an ESCALAB 250Xi X-ray photoelectron spectrometer with the Mg K$\alpha$ radiation. The morphology information of photocatalysts was recorded on scanning electron microscope (SEM) using a Nova NanoSEM 450. Ultraviolet-visible

diffuse reflectance spectra (UV-vis DRS) of as-prepared samples were recorded on a UV-vis spectrophotometer (Shimadzu UV-3600PLUS) with a range from 200 to 800 nm using $BaSO_4$ as a reflectance standard material. The photoluminescence spectra (PL) of the photocatalyst were measured with a Varian Cary Eclipse spectrometer.

The electrochemical analyzer (CHI660B) was used to measure the photoelectrochemical performance of the photocatalysts. This equipment contained a three-electrode system of platinum wire, saturated Ag/AgCl electrode and $1 \times 1$ cm$^2$ $Ag_3PO_4$/ITO glass as the working electrode. Photocurrent measurement was carried out in 0.1 M phosphate-buffered saline under visible light (pH = 7.0), while the electrochemical impedance spectroscopy (EIS) was performed in a 0.1 M KCl solution containing 5mM Fe $(CN)6^{3-}$/Fe $(CN)6^{4-}$.

*3.4. Photocatalytic Activity*

In order to determine the photocatalytic activity of $Ag_3PO_4$ photocatalysts, the photodegradation of RhB and 4-CP molecules were carried out under visible light irradiation (250 W Xe lamp with a 400 nm cut-off filter). The 20.0 mg photocatalyst was dispersed into 100 mL aqueous solution of RhB and 4-CP (10 mg/L) in a Pyrex photocatalytic reactor, respectively. The mixed solution was stirred for 30 min under dark conditions to reach the adsorption-desorption equilibrium. After irradiation, 4 mL suspension was sampled within a certain irradiation time interval and then centrifuged to remove the photocatalyst particle. For RhB photodegradation, the light absorbance of the resulted solution was determined by UV-Vis spectrophotometer (TU-1810PC). The concentration of 4-CP solution was detected on an Agilent high performance liquid chromatography (HPLC, Agilent LC1200) with C18 column. Mobile phase was the miscible liquids of 70% methyl alcohol and 30% pure water (V/V), and the flow rate was confined as 1.0 mL/min with a detection wavelength of 225 nm.

A free radical trapping experiment was carried out to study the photodegradation mechanism. All trapping experimental processes were the same as that RhB photodegradation except for the addition of different scavengers of ethylenediaminetetraacetic acid disodium salt (EDTA-2Na, 5 mM), p-benzoquinone (2 mM), and tert-butanol (t-BuOH, 5 mM).

## 4. Conclusions

In summary, the spherical-like IL-$Ag_3PO_4$ photocatalyst with exposed {111} facets has been successfully synthesized via the chemical precipitation method in the presence of reactive ionic liquid [Omim]$H_2PO_4$. The presence of exposed {111} facet was advantageous for facilitating the charge carrier transfer and separation. The resulted IL-$Ag_3PO_4$ sample exhibited highly photocatalytic activity for the degradation of RhB and 4-CP molecules under visible light irradiation. The photodegradation rates for the removal of RhB and 4-CP molecules were 98.4% and 68.23% within 180 min over IL-$Ag_3PO_4$ sample. The efficiencies of IL-$Ag_3PO_4$ material for RhB and 4-CP photodegradation were 1.94 and 2.45 times higher than that of $Ag_3PO_4$, respectively. The enhanced photocatalytic activity was mainly attributed to the synergistic effects of the exposed {111} facets and enhanced photoabsorption. The result of radical trapping experiment demonstrated that the active species of holes and •$O_2^-$ played a vital role in RhB photodegradation under visible light over IL-$Ag_3PO_4$ photocatalyst. This study will provide a promising prospect for ionic liquid-assisted synthesis of photocatalysts with a high efficiency.

**Author Contributions:** Writing—original draft, Methodology, Investigation, B.Z.; Software, visualization, L.Z.; Data curation, formal analysis, Y.Z.; Funding acquisition, Writing—review & editing, project administration, C.L.; Conceptualization, resources, J.X.; Supervision, validation, H.L. All authors have read and agreed to the published version of the manuscript.

**Funding:** This research was funded by the National Natural Science Foundation of China (No. 51902282), and Qinglan Project of Jiangsu Province of China.

**Data Availability Statement:** Not applicable.

**Conflicts of Interest:** The authors declare no conflict of interest.

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
