# Peer review of "Ionic Liquid-Assisted Synthesis of Ag3PO4 Spheres for Boosting Photodegradation Activity under Visible Light"

_catalysts, doi:10.3390/catal11070788_

Round 1
Reviewer 1 Report
The authors presented a study on photocatalytic activity of Ag3PO4 and IL-Ag3PO4. The authors characterized the synthesized Ag3PO4 with variety of methods such as XRD, SEM, XPS and so on. The authors also tested the photocatalytic performance by pollutant decomposition. Overall, the manuscript is OK, however, it can be improved and my suggestions are as follows:
1. Pg2 line 52 “The presence of {111} facets in Ag3PO4 crystal has a higher barrier, which could hinder the electron transfer and the charge carrier recombination, resulting in the efficient mobility/separation of charge carrier on the facets and thus the highly photocatalytic performance” How can the photocatalytic performance be enhanced when electron transfer is hindered? There maybe a fundamental problem here since photocatalytic activity heavily relies on electron/hole transfer ability.
2. Fig. 3a, why there is a baseline for DRS spectra for IL-Ag3PO4
3. Fig. 3b, is Ag3PO4 direct or indirect gap semiconductor? Why use (have)^1/2?
Reviewer 2 Report
In this paper, the authors reported on the synthesis and characterization of the spherical-like Ag3PO4 material (IL-Ag3PO4) with exposed {111} facet in the presence of reactive ionic liquid 1-butyl-3-methylimidazole dihydrogen phosphate ([Omim]H2PO4). Although the proposed synthesis method is unique and the paper includes some interesting results, the manuscript needs major revision prior to being published. The following points should be addressed in the revision;
1) Overall, English needs to be improved.
2) Based on the SEM observation results, the IL-Ag3PO4 has significantly larger particle size than that of the Ag3PO4. In addition, in the case of the IL-Ag3PO4, Ag3PO4 particles are embedded in the IL. These mean that the substrate such as rhodamine B (RhB) and 22 p-chlorophenol (4-CP) need to diffuse in the IL layer to reach to the Ag3PO4 particles. This would cause high mass transfer resistance for the IL-Ag3PO4. Besides, since the IL-Ag3PO4 has visible light absorption property, it causes self-shielding of the Ag3PO4 particles. Because of the self-shielding effect and larger particle size, it is possible that the Ag3PO4 particles located inside the IL-Ag3PO4 cannot be fully utilized. In spite of these disadvantageous properties, why the IL-Ag3PO4 can exhibit superior photocatalytic performance than the conventional Ag3PO4.
3) Do you have any evidence that the adsorption of rhodamine B (RhB) and 22 p-chlorophenol (4-CP) obviously reached equilibrium before photocatalytic degradation experiment?
4) In Figure 5(b), new two peaks are observed other than 4-CP peak. However, how can you tell the new two peaks are from 4-CP degradation? Since in the case of the IL-Ag3PO4, Ag3PO4 particles are embedded in the IL, the contact between the Ag3PO4 particles and the IL should be more frequent than that between Ag3PO4 particles and the 4-CP. It means that degradation of IL by the Ag3PO4 particles could proceed more than the degradation of the 4-CP. If that is the case, the two new peaks in the HPLC spectra could be caused by the degradation intermediate of the IL and the decrease of the 4-CP in Fig.5(a) is caused by merely the adsorption of the 4-CP in the sample.
5) In L154-157, the authors explained that “an intensity ratio of {222}/{200} facets of IL-Ag3PO4 (1.06) are significantly higher 155 than that of Ag3PO4 crystal (0.83), which showed that the surface of IL-Ag3PO4 crystal is mainly {111} facets”. However, as the authors cited in the ref.16, Ag3PO4 tetrahedrons showed an intensity ratio of 1.72 between {222} and {200} planes, while Ag3PO4 cubes and Ag3PO4 spheres with mixed facets had an intensity ratio of 0.951 and 1.23 respectively, confirming that Ag3PO4 tetrahedrons were mainly composed of {111} crystalline planes. The value of the ratio of {222}/{200}, 1.06 reported by the authors is much closer to the value of cubic or spherical Ag3PO4 than Ag3PO4 tetrahedrons which were mainly composed of {111} crystalline planes. That means, the IL-Ag3PO4 sample could have less {111} facet than even the conventional spherical Ag3PO4 and have similar contribution of {111} facet to the cubic Ag3PO4. This also means that the IL-Ag3PO4 could have lower photocatalytic activity than the conventional spherical or cubic Ag3PO4 due to its low {222}/{200} ratio, large particle size, and possible high mass transfer resistance (caused by IL layer).
6) The authors mentioned about the enhanced photoabsorption for the IL-Ag3PO4 in the manuscript. Is there any experimental result to support this?
7) What is the advantage of the proposed method compared to the conventional synthesis method of Ag3PO4 tetrahedrons?

Round 2
Reviewer 2 Report
Explanations by the authors in their response to my review comments are totally not convincing. Still, I do not understand why the author's photocatalyst could show high photocatalytic activity. However, the results are interesting and worthwhile for publication.